# Enhancement of Stability in *n*-Channel OFETs by Modulating Polymeric Dielectric

**DOI:** 10.3390/polym15112421

**Published:** 2023-05-23

**Authors:** Po-Hsiang Fang, Peng-Lin Kuo, Yu-Wu Wang, Horng-Long Cheng, Wei-Yang Chou

**Affiliations:** 1Department of Photonics, National Cheng Kung University, Tainan 70101, Taiwan; 2Graduate Institute of Photonics, National Changhua University of Education, Changhua 50007, Taiwan

**Keywords:** polymer, organic field-effect transistors, memory devices, electric dipoles, PTCDI-C_13_, stability

## Abstract

In this study, a high-K material, aluminum oxide (AlO*_x_*), as the dielectric of organic field-effect transistors (OFETs) was used to reduce the threshold and operating voltages, while focusing on achieving high-electrical-stability OFETs and retention in OFET-based memory devices. To achieve this, we modified the gate dielectric of OFETs using polyimide (PI) with different solid contents to tune the properties and reduce the trap state density of the gate dielectric, leading to controllable stability in the N, N’-ditridecylperylene-3,4,9,10-tetracarboxylic diimide (PTCDI-C_13_)-based OFETs. Thus, gate field-induced stress can be compensated for by the carriers accumulated due to the dipole field created by electric dipoles within the PI layer, thereby improving the OFET’s performance and stability. Moreover, if the OFET is modified by PI with different solid contents, it can operate more stably under fixed gate bias stress over time than the device with AlO*_x_* as the dielectric layer only can. Furthermore, the OFET-based memory devices with PI film showed good memory retention and durability. In summary, we successfully fabricated a low-voltage operating and stable OFET and an organic memory device in which the memory window has potential for industrial production.

## 1. Introduction

Organic semiconductor materials are widely used in optoelectronics, such as organic field-effect transistors (OFETs) [1,2], organic solar cells [3,4], organic light-emitting diodes (OLEDs) [5,6], and organic volatile/non-volatile memory devices [7,8]. The advantages such as low power consumption, low cost, light and thin products, and easy recycling of organic semiconductor devices are in line with the trend of environmental protection and have become the focus of current development [9,10]. However, in organic electronic devices, the research on OFETs is the most critical, because OFETs are the basic unit elements in the design of large-scale integrated circuits. The index characteristic of the pros and cons of an OFET is carrier mobility. In addition, various factors affect the carrier transport speed in the channel of an OFET, such as the stacking degree of organic semiconductor molecules, the ordering degree of molecular arrangement, the density of defect states at the semiconductor/dielectric layer’s interface, the amount of carrier injection energy barriers, the degree of energy band matching, the impedance value between interfaces, and so on [11,12,13,14,15,16,17,18,19]. Among them, the characteristics of the interface between the dielectric layer and the semiconductor layer have a great effect on the carrier transport of the overall device. Meanwhile, in the past, various research groups used small molecule self-assembled layers (SAM) to reduce the density of defect states on the surface of the dielectric layer or to improve the crystallinity of the semiconductor layer by changing the hydrophilic/hydrophobic properties of the surface of the dielectric layer [20,21,22,23]. The polymer dielectric materials commonly used in research include poly (4-vinyl phenol) (PVP), poly (methyl methacrylate) (PMMA), polystyrene (PS), polyimide (PI), and so on. The dielectric material PI has good mechanical properties, good thermal stability, and good insulating properties and is easy to be modified via processing. OFET-based memory devices, in which a charge-storing electret layer of PI thin films is inserted between the semiconductor and insulating layers, have been extensively studied [24,25,26]. Liu et al. used polymer blends to fabricate a high-performance OFET with non-volatile memory (ONVM) based on pentacene. From the research results, the polymer blends significantly improve the memory performance of pentacene-based memory devices [27]. Based on the research, the PI dielectric layer material has electric dipoles in its structural branches, which can enhance the ability of the field effect, thereby enhancing the ability to trap the carriers of the organic semiconductor layer and enhance the storage characteristics of the memory device [28]. However, a stability issue in OFETs is related to gate bias stress, which refers to the long-term exposure of the device to a constant gate voltage [29,30,31]. This can lead to the accumulation of charges in the dielectric layer, which can affect carrier mobility and lead to a degradation in device performance. In particular, the stability of the PI dielectric layer material is affected by gate bias stress, which is caused by the presence of a quasi-permanent dipole field within the molecule. This can attract charged carriers in the channel, causing electrical hysteresis and reducing the current of the transistor device under long-term operation. Therefore, our research direction is to explore how to optimize the PI dielectric layer material to improve the stability of an OFET.

In this study, aluminum oxide (AlO*_x_*) and N, N’-ditridecylperylene-3,4,9,10-tetracarboxylic diimide (PTCDI-C_13_) are used as the dielectric layer and organic semiconductor layer of OFETs, respectively. In addition, a PI film is coated between the dielectric and semiconductor layers to fabricate low-voltage-driven transistor devices. Hence, by changing the solid content of PI to control its surface, physical properties and the carrier capture and release behaviors at the interface between PI films and semiconductors were investigated. Using this novel PI/AlO*_x_* dielectric material, OFETs were fabricated with high stability. In addition, the physical mechanism of OFETs with high stability is explored in detail in this paper. Moreover, this dielectric can also be applied to organic memory devices to understand the relationship between the carrier capture and release behaviors of PI with different solid contents and memory characteristics.

## 2. Materials and Methods

### 2.1. Device Fabrication

The structure of the OFET-based device is shown in Figure 1a. The area of the glass substrate of the OFET was 2 cm × 1.5 cm. Meanwhile, an aluminum film was deposited on a glass substrate to act as the gate electrode, and then AlO*_x_* was formed on the surface of aluminum treated with oxygen plasma. Afterward, PI (from Daxin Materials Corporation, Taichung City, Taiwan, Appendix A) and *n*-methylpyrrolidone (NMP) were prepared into solutions based on the content of PI, and the solid content of PI in the solution was 6.4 wt%, 4.7 wt%, 3.2 wt%, and 1.6 wt%. Then, these solutions were spin-coated onto the AlO*_x_* and baked at 220 °C for 1 h to form the PI/AlO*_x_* dielectric layer, in which the thicknesses of the PI films were 136, 96, 55, and 24 nm. Subsequently, the 40 nm thick PTCDI-C_13_ film (purchased from Sigma Aldrich, Saint Louis, MO, USA, Appendix A) as an active layer of the OFET was thermally grown on the PI/AlO*_x_* dielectric layer at a deposition rate of 0.2 Å/s and a base pressure of 3 × 10^−6^ Torr. Then, after depositing the active layers, 80 nm thick Ag as the source and drain electrode was deposited on the active layer through a shadow mask with a channel length (*L*) and width (*W*) of 100 and 2000 μm, respectively.

### 2.2. Characterization

In this study, a coplanar metal–insulator–metal (MIM) structure was used to study the capacitor of the dielectric layer, and the capacitance–voltage (C-V) was measured using the Agilent E4980A LCR meter at a measurement frequency of 1 kHz and an operating voltage from −2 V to 2 V. AlO*_x_* is a high-dielectric material; thus, it exhibited an ultra-high capacitance value of 1007.1 nF/cm^2^ in the capacitance measurement. With the increase in the solid content of PI, the capacitance value of the PI/AlO*x* dielectric layer gradually decreased, and the capacitance values were 234.41, 72.91, 27.91, and 19.67 nF/cm^2^, as shown in Figure 1b. In addition, the electrical characteristics of the OFETs were measured using a Keithley 4200 semiconductor characterization system, and the memory properties of the OFET-based memory devices were measured using a Keithley 2636 semiconductor characterization system within a nitrogen-filled glove box. The surface morphologies and optical properties of all thin films in PTCDI-C_13_-based OFETs were detected via atomic force microscopy (AFM, Park XE-100, Park Systems Corp., Suwon, Republic of Korea), X-ray diffraction (XRD) with a Cuka_1+2_ of λ = 1.54184 Å (Bruker D8 Discover, Bruker, Billerica, MA, USA) and a UV-visible absorption spectrometer (GBC Cintra 202 UV–Vis spectrometer with a resolution of less than 0.9 nm, GBC, Lake Zurich, IL, USA).

## 3. Results and Discussions

### 3.1. Surface Morphology Analyses of Dielectric Layers

First, the surface morphology and surface roughness of the dielectric layer was observed with an AFM in a scanning range of 5 × 5 μm^2^, as shown in Figure 2. When the solid content of PI was 6.4 wt%, the surface morphology of the PI/AlO*_x_* dielectric layer was smooth, and its surface root mean square roughness (*R*_q_) was 0.947 nm. Moreover, as the NMP diluted PI, the solid content of PI gradually decreased from 4.7 wt% to 1.6 wt%, thereby resulting in various granular grains starting to appear on the surface of the PI/AlO*_x_* dielectric layer. The PI/AlO*_x_* dielectric layers with a lower solid content of PI, 4.7%, 3.2%, and 1.6%, had more grains and rougher surfaces, with a *R*_q_s of 2.612 nm, 4.788 nm, and 7.325 nm, respectively, because of the various pin holes. However, the AlO*_x_* dielectric layer not covered with PI had more grains on its surface and was the roughest, with a *R*_q_ of 8.668 nm. From the experimental results, when a lower solid content of PI is formed on the AlO*_x_* dielectric layer, the surface roughness increases with the decrease in the solid content of PI. Considering that the thickness of the PI films was thinner, the surface roughness can be affected by various tiny pores on the surface of the PI/AlO*_x_* dielectric layer, thereby resulting in a rougher surface and a larger *R*_q_ value.

### 3.2. Analysis of Thin Film Microstructure

Herein, the molecular stacking and film microstructure of the PTCDI-C_13_ film grown on the PI film were verified via UV-Vis absorption and XRD characteristics. Organic small-molecule materials have π electrons that absorb ultraviolet light to excite electrons to higher energy levels, which are influenced by molecular crystallization. This phenomenon can be used to analyze the crystallinity of thin films. Figure 3a shows the normalized absorption spectra of PTCDI-C_13_ films grown on PI films with different solid contents and AlO*_x_* dielectric layers. It can be observed that the absorption peaks of the PTCDI-C_13_ films are located at wavelengths of 488.5 nm, 527.5 nm, and 571.3 nm. The absorption spectral bands of these PTCDI-C_13_ films are in the range of green light, which is consistent with the results in the literature. Meanwhile, from the experimental results, PTCDI-C_13_ films grown on PI films with different solid contents have wider spectral distribution and stronger absorbance those that grown on AlO*_x_* dielectric layers only. Moreover, the absorption peaks above 486 nm of the UV-Vis spectra of PTCDI-C_13_ films grown on PI films with different solid contents have a more redshift, and the absorbance is higher than that of PTCDI-C_13_ film grown on the AlO*_x_* dielectric layer. Therefore, it can be speculated from the UV-Vis absorption that the PTCDI-C_13_ film grown on the PI surface will obtain a better molecular stacking arrangement.

Meanwhile, XRD analysis was used to measure the changes in molecular crystal arrangement of PTCDI-C_13_ films grown on PI films with different solid contents and native AlO*_x_* film. From the comparison of XRD spectra in Figure 3b, the (001) peak of the PTCDI-C_13_ film angle is located at 3.33°. However, as the solid content of PI decreases, the intensity of the (001) peak also decreases; particularly, the PICDI-C_13_ film grown on the native AlO*_x_* film has the lowest peak intensity. Combining the results in Figure 2 to compare all the PTCDI-C_13_ films grown on the surface of the dielectric layer, the intensity of the diffraction peaks tends to decrease with the increase in the surface roughness of the dielectric layer. However, at the (001) peak, the FWHM of the PTCDI-C_13_ film grown on the PI layer with a solid content of 6.4 wt% is 0.242°, which is smaller than that grown in other PI films with different solid contents, thereby indicating better crystal quality. Moreover, a (002) peak is also observed at the angle of 6.73°, which indicates that the surface of the modification layer is smooth to make the PTCDI-C_13_ film grow better. Thus, based on the analysis results of UV-Vis absorption and XRD, the flatter the surface of the dielectric layer becomes, the better the PTCDI-C_13_ crystallinity is.

### 3.3. Electrical Measurements of OFETs

Herein, the electrical performances of OFETs based on the dielectrics with various solid contents of PI are investigated in sequence. The transfer and output characteristics of these devices are shown in Figure 4. It was observed that the electrical properties of the OFETs have a very significant correlation with the dielectric layer, including the threshold voltage (*V*_th_), on-off ratio (*I*_on_/*I*_off_) and subthreshold swing (*S.S.*) [32], where $S.S.=\partial V_{\mathrm{GS}}/\partial \log\left(I_{\mathrm{DS}}\right)$ is defined as the change in the gate voltage (*V*_GS_) required to increase the drain current (*I*_DS_) by one decade. The detailed electrical parameters of five types of OFETs are listed in Table 1 (Appendix A shows the average and standard deviation of the electrical characteristics using 10 devices.). Meanwhile, the transfer characteristics of the PI- and AlO*_x_*-based OFETs measured in the glove box are shown in Figure 4a–e, respectively, in which the gate-source voltage (*V*_GS_) was swept from −2.0 V to 2.0 V, and the drain voltage (*V*_DS_) was kept at 2.0 V. As the solid content of the PI in the PI/AlO*_x_* dielectric layer decreased, the saturated *I*_DS_ of the transfer curve significantly increased, but the off current (*I*_off_) almost maintained unchanged, thereby indicating that the field effect was enhanced by the decrease in the thickness of PI film. As shown in Figure 4a, when the solid content of the PI film is 6.4 wt%, the on/off ratio is 7.0 × 10^2^, and when the solid content of the PI film decreases to 1.6 wt%, the on/off ratio increases to 6.5 × 10^3^, as shown in Figure 4d, because of the enhancement of *I*_DS_. The on/off ratio of the device with AlO*_x_* dielectric only is 4.8 × 10^1^, as shown in Figure 4e, caused by a significant increase in *I*_off_. However, the electrical hysteresis of the OFET with the AlO*_x_* dielectric layer only is higher than that of other OFETs with PI/AlO*_x_* dielectric layers. Meanwhile, from the analysis of AFM images, the surface of the AlO*_x_* dielectric layer is rough, resulting in the generation of various trapped states; hence, there will be hysteresis. The trap state density, *N_SS_*, which includes the bulk trap density of organic semiconductor and the interface trap density between the organic semiconductor and dielectric, was calculated to realize the mechanism of electrical hysteresis from the value of *S.S.* using the following Equation (1) [33]:(1)$$N_{\mathrm{SS}}=\left(\frac{S.S.log(e)}{kT/q}-1\right)\frac{C}{q}$$
where *N_SS_* is the defect density of states, *S.S.* is the subthreshold swing, and e = 2.71828 is the Euler constant. As shown in Figure 5, the *S.S.* value of the OFET whose dielectric layer is AlO*_x_* only at the beginning is 2.18 V/dec and the *N_SS_* is the largest. When the solid content of the PI film is increased to 3.2 wt%, the *S.S.* is the smallest among all OFETs, and the *N_SS_* is 1.3 × 10^12^ cm^−2^ eV^−1^ (Appendix A shows the relationship between the *S.S.* and *N_SS_* versus the different solid contents of PI layers by averaging 10 devices). Afterward, although the *S.S.* will gradually deteriorate, the higher the solid content of the PI film, the saturation phenomenon is observed for the *N_SS_*. In addition, in the output curve part, the saturation current value of the PI film with a solid content of 6.4 wt% is only 4 nA, as shown in Figure 4f. Herein, the output saturation current increases with the decrease in the solid content of the PI film, and when the solid content of the PI film is 1.6 wt%, the output saturation current reaches a maximum value of 162 nA. The field effect strength of the gate voltage is related to the thickness of the overall device; hence, the lower the thickness of the dielectric layer, the stronger the field effect ability, which can accumulate more carriers in the organic semiconductor layer to increase the channel current. Therefore, even though the better crystallinity of PTCDI-C_13_ was induced by a smoother surface of the PI 6.4 wt% dielectric layer, a thicker PI film may result in a weaker gate field for achieving lower-charge carrier mobility than that with the thinner PI dielectric layers. The field-effect mobility of electronics can be extracted from the *I*_DS_ in the saturation regime of the transfer characteristics as follows in Equation (2) [34]:(2)$$I_{\mathrm{DS}}=\frac{1}{2}\mu C\frac{W}{L}(V_{\mathrm{GS}}-V_{\mathrm{th}})$$
where *μ* is the field-effect mobility, *C* is the capacitance per unit area of the gate dielectric, *W* is the channel width, and *L* is the channel length. Therefore, the better carrier mobility of 1.58 × 10^−2^ cm^2^/Vs can be obtained from the OFET with the PI film in which the solid content is 1.6 wt%. To highlight the performances of our devices, we compare our device to low-voltage operating organic transistors reported in the literature over the past decade. The corresponding electrical characteristics of our OFETs operating at 2 V are excellent, as shown in Appendix A [35,36,37,38,39,40,41,42].

### 3.4. Stability Analysis of OFETs

The time-dependent measurements of electrical characteristics were down for all OFETs to realize the effect of a dielectric layer on operation stability. The operation stabilities of the devices were measured by applying the constant bias voltages of *V*_GS_ = 2 V and *V*_DS_ = 2 V to the transistors for a long time, as shown in Figure 6a. Herein, when the PI solid contents were 6.4 wt% and 4.7 wt%, the *I*_DS_ of the devices increased with the increased in the measuring time. Thus, it is speculated that when the solid content concentration of the PI film is high, the carriers in the device channel are still accumulated under continuous operation to cause a continuous increase in the channel current. The accumulation of carriers over time was mainly because of the dipole field, which was derived from the electric dipoles aligned by the gate field in the PI film [31,43,44,45]. When the solid contents of the PI films were down to 3.2 wt% and 1.6 wt%, the channel currents of the devices decayed with time; however, the channel current tended to saturate after the continuous measurement time exceeded 1000 s. This means that the accumulation of carriers induced by the dipole field in lower PI solid contents is insufficient to compensate the trap states induced by the gate field. In particular, the OFET with the AlO*_x_* dielectric layer only cannot be operated for a long time, and the collapse phenomenon occurs mainly because of the breakdown of the dielectric layer at 2000 s. An analysis of the change in the drain current with time ($\partial I_{\mathrm{DS}}/\partial t$) was performed, as shown in Figure 6a, to study the relationship between the stability and the solid content of PI for all OFETs, as shown in Figure 6b. To assess the stability of each PI device, we differentiated the drain current with respect to time. The change of drain current over time for OFETs with the solid continent of 3.2 wt% and 1.6 wt% in PI thin film was almost zero after 1000 s, exhibiting the most stable performance. In contrast, the $\partial I_{\mathrm{DS}}/\partial t$ slowly decayed over time until 5000 s after the measurement was still not zero for the OFETs with the solid continent of 4.7 wt% and 6.4 wt% in PI films, as depicted in Figure 6b, indicating that slight instability was observed.

We applied a fixed bias, *V*_GS_ = 2 V, and *V*_DS_ = 2 V, to the organic device for 300 s, 1800 s, 3600 s, and 7200 s to study the variation in *N_SS_* to understand the effect of defect density of the dielectric layer for the OFETs under long-term operation, as shown in Figure 7. If we only use AlO*_x_* as the dielectric layer in OFET, the *N_SS_* of the device increases with the increase in the measurement time. The initial *N_SS_* value is 3.0 × 10^13^ cm^−2^ eV^−1^, and then its maximum value is 1.84 × 10^14^ cm^−2^ eV^−1^, thereby indicating that the use of inorganic materials alone as the dielectric layer of the OFET will result in a substantial increase in the gate field-induced defect state density with an increasing operating time. When the device adopts AlO*_x_*/PI (1.6 wt%) as the dielectric layer, the *N_SS_* decreases to 6 × 10^12^ cm^−2^ eV^−1^; however, the value of *N_SS_* slightly increases with the measurement time, and then it is kept almost the same after 2000 s. This is what causes the drain current to decay with time and saturate after 2000 s. A similar phenomenon also occurs in the device with an AlO*_x_*/PI (3.2 wt%) dielectric layer. Furthermore, if the solid content of PI in the dielectric layer of the device further increases, such as in the case of a 4.7 wt% and 6.4 wt%, the value of *N_SS_* decreases slightly over time. Meanwhile, when the solid content of PI in the dielectric layer is 6.4 wt%, the value of *N_SS_* is at least about 5 × 10^11^ cm^−2^ eV^−1^. Based on the above experimental results, there is the largest density of defect states in the OFET using inorganic AlO*_x_* as the dielectric layer. Various defect states continuously capture carriers during the operation of the device, which reduces the drain current of the device. On the contrary, when the PI film with high solid content acts as the modification layer on the AlO*_x_* layer, there are fewer defect states, and the gate field-induced defect states are compensated for by the carriers induced by the dipole field created by the dipoles within the PI film during the operation of the device, so that carriers can be accumulated to increase the drain current with time.

### 3.5. Analyses of Memory Device Characteristics

The electrical stability decreases with the decrease in the solid content of PI in the PI/AlO*_x_* dielectric layer of the OFET, thereby indicating that the trapping capability of carriers increases with a decreasing solid content of PI. However, a higher trapping capability does facilitate the writing process in organic memory devices. The advantage of the fast response of electric dipoles in PI to electric fields helps the charge erasing process of organic memory. Therefore, an excellent OFET-based memory device fabricated with a low solid content of PI was expected. A positive pulse voltage (0.5 to 3 V, pulse time of 1 s) was applied to the gate of the OFET. The gate voltage and the electric field generated by the electret captures the electrons in the organic semiconductor to the interface between the PI dielectric layer and the organic semiconductor. At this time, the first transistor transfer curve is measured, and the threshold voltage obtained after the data processing is the write voltage. Next, a negative pulse voltage (−0.5 to −3 V, pulse time of 1 s) is applied to the gate of the OFET. The captured electrons within trapping states at the interface between the dielectric layer and the organic semiconductor are released, hence, the threshold voltage is shifted again, and the erasing voltage is obtained. Finally, the writing and the erasing voltages are subtracted to obtain the memory window, which represents the quality of the memory characteristics for the device, as shown in Figure 8a. When the PI film with a solid content of 6.4 wt% was given a pair of pulse voltages of writing of 0.5 V and erasing of −0.5 V, the memory window was 0 V, and as the pulse voltage increased to 3 V, the memory window also increased to 0.13 V. However, when the solid content of the PI film decreased from 4.7 wt% to 1.6 wt%, the memory windows biased at the pulse voltage of ±3 V were 0.15 V, 0.31 V, and 0.54 V; the raw data of the memory device is shown in Appendix A. The OFET with a solid content of 1.6 wt% PI is an excellent candidate for a memory device. This OFET-based memory device has a memory window of over 0.5 V under the condition that low voltage pulses of 3 V and −3 V are applied during the writing and erasing processes, respectively. Memory windows greater than 0.5 V can already be used meaningfully in circuit simulations and real-world devices. When the dielectric layer of the memory device is AlO*_x_* only, the memory window biased at the pulse voltage of ±3 V is 0.03 V. Based on the experimental results, the increase in the memory window is mainly because of the significant decrease in the thickness of the carrier trapping layer, in which the trap ability of the gate bias is enhanced by increasing the pulse voltage. However, the memory window of the device with PI (1.6 wt%)/ AlO*_x_* dielectric is larger than that with an AlO*_x_* dielectric only. This result is because the electric dipole in PI can quickly react to the external gate field, thus resulting in the easy capture and release of carriers during programming and erasing processes, respectively.

Memory retention refers to the time that data can be stored and preserved when data are written into the device during the operation of the memory device. Herein, we measured the characteristic curve of *V*_GS_-*I*_DS_ to calculate the threshold voltage, and then the *V*_GS_-*I*_DS_ curve was measured again every 30 min. The total measurement time was 120 min to record the results of the measured *V*_th_ at each time, as shown in Figure 8b. After one data writing and saving operation, the memory effect significantly decreases over time for the memory device only using an AlO*_x_* material as the dielectric layer. For the three memory devices with the PI films whose solid contents range from 1.6 wt% to 4.7 wt%, the *V*_th_ of the memory devices all gradually decrease within 30 min after one data writing and storage operation, thereby indicating that it also represents a loss of data. However, after the measurement time exceeds 30 min, the value of *V*_th_ gradually stabilizes. Then, after the memory device with a PI film of solid content of 6.4 wt% has been written and stored once, the *V*_th_ of the memory device remains almost unchanged over a long time of measurement. Hence, it is speculated that because the thickness of the carrier capture layer is too thick, weak data can be written. However, the capture ability of the carrier capture layer is not strong; that is, it has a low trap state density and the carrier cannot be captured for a long time, thus resulting in a slow loss of data. Based on the experimental results, if there is a low solid content PI layer in the structure of a memory device, the device will have a larger memory window and stronger retention ability.

In addition, memory durability represents how many times the memory device can withstand repeated operations of data writing and erasing without losing data. Figure 9 shows the results of the memory endurance of memory devices with different dielectric layers. The interactive change in the writing voltage (3 V, pulse time of 1 s) and erasing voltage (−3 V, pulse time of 1 s) is used to measure the change in the memory window over time. Hence, it is shown that the memory window of the device with PI films of different solid contents can maintain a horizontal state without decreasing the memory window as the number of operations increases. The memory device with a solid content of 1.6 wt% PI still maintains a memory window of above 0.59 V after the memory endurance analysis of 50 cycles, so this device is a memory device with excellent stability. The research findings indicate that the write–erase ability is optimized when the solid content of PI in the dielectric layer is 1.6 wt% or 3.2 wt% for the memory devices. The *V*_th_ variation of each storage device in write and erase cycles is shown in Appendix A.

The charge carrier density (Δ*n*) stored in memory devices during the writing and erasing processes was calculated using the following formula [46]:(3)$$∆n=\frac{∆V_{\mathrm{th}}\times C_{i}}{e}$$
where *e* is the amount of elementary charge (1.602 × 10^−19^ C), and C*_i_* is the capacitance per unit area. The value of Δ*n* determines the strength of the built-in electric field caused by a charge injection and reflects the quality of the storage performance directly. After 50 write-erase cycles, the charge storage density of the PI dielectric with a solid content of 1.6 wt% is the highest, resulting in the largest memory window among all memory devices. This means that the use of PI with a solid content of 1.6 wt% as a dielectric layer in organic memory devices can exhibit a high charge storage capacity and memory stability. However, when the solid content of PI is increased beyond this optimal level, the charge carrier density decreases, resulting in a reduced memory window, as displayed in Table 2.

However, for memory devices with AlO*_x_* dielectric layers only, the memory window varies drastically with the number of operations. Therefore, it is determined that using PI as the dielectric layer, its memory device can maintain a stable memory window under repeated operations, and has good, repeated reading and writing abilities.

## 4. Conclusions

We diluted PI with NMP to form a series of PI films on the surface of AlO*_x_*, and the solid contents of the PI films were formulated as 6.4 wt%, 4.7 wt%, 3.2 wt%, and 1.6 wt% to act as the dielectric of OFETs. When the solid content of the PI film gradually decreases, the thickness of the carrier trapping layer becomes thinner and the surface polarity term increases; thus, the field effect capability is enhanced to make the organic semiconductor easily accumulate carriers and increase the channel current of the device. However, s decrease in the solid content of the PI film will increase the surface roughness, thereby resulting in poor crystal growth quality of PTCDI-C_13_, and thus affecting the carrier transport of the organic semiconductor layer and reducing the carrier mobility. Meanwhile, from the XRD experimental results, the organic semiconductor material grown on the PI film with a solid content of 6.4 wt% has good crystalline performance. However, the performance of electrical characteristics is poor because the thickness of the dielectric layer is thick to reduce the gate field. Moreover, based on the experimental results of the capacitance value, the lower the solid content of the PI film, the greater the capacitance value. A calculation of the interfacial defect state density was carried out for the OFETs with different dielectric layers from the capacitance value and the subthreshold swing of the transistor, resulting in a relationship whereby the lower the solid content of the PI film, the higher the interfacial defect energy state density. Therefore, in the case of measuring drain current–time, a phenomenon of current growth over time was observed in the device with a high solid content PI film (6.4 wt% and 4.7 wt%) because of gate-induced trap states compensated for by the accumulation carriers induced by the electric dipoles within the PI film. Meanwhile, in cases with a lower solid content of PI film and native AlO*_x_* film, the carriers in the organic semiconductor need to continuously fill the trapped states induced by the gate field and poor crystalline PTCDI-C_13_, its drain current decreases with time, and after it is gradually filled, the drain current value tends to gradually stabilize, indicating that stable electrical characteristics are achieved. Furthermore, the electrical stability of OFETs can be controlled according to the controllable trap state density from the gate dielectric material. The electrical stability that can be achieved in the case with a low solid content of PI is the case for making excellent organic memory devices. From the experimental results of the memory device, the memory device with the PI layer with different solid contents can still maintain a good memory window after multiple writing and erasing operations. In most transistors, including organic and inorganic ones, the electrical stability decays with operation time; however, in our experiments, organic transistors can show excellent stability. Thus, these high-stability devices will meet the consumer demand for flexible electronic products in the future.

## Figures and Tables

**Figure 1 polymers-15-02421-f001:**
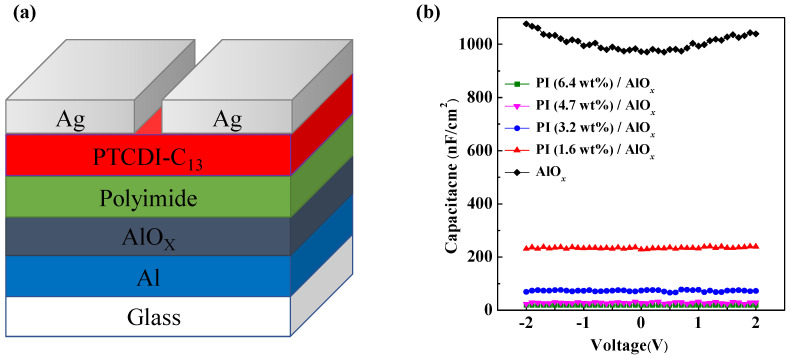
(**a**) Architecture of a PTCDI-C_13_-based OFET; (**b**) capacitance of various dielectric layers measured at a frequency of 1 kHz.

**Figure 2 polymers-15-02421-f002:**
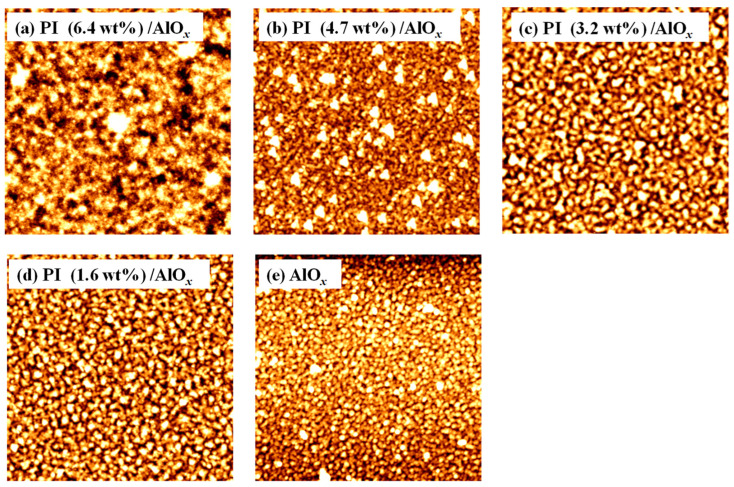
AFM surface morphologies of different kinds of dielectric layers; the solid content of PI films of PI/AlO*_x_* dielectric layers are (**a**) 6.4 wt%, (**b**) 4.7 wt%, (**c**) 3.2 wt%, and (**d**) 1.6 wt%, (**e**) AlO*_x_* dielectric layer.

**Figure 3 polymers-15-02421-f003:**
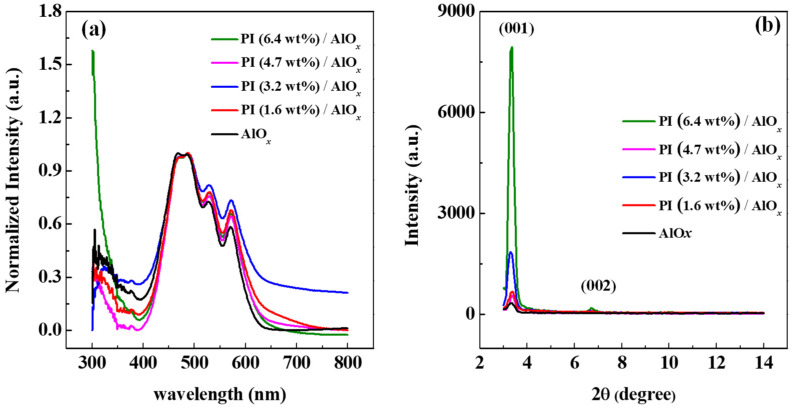
(**a**) Absorption spectra normalized at wavelength of 500 nm and (**b**) X-ray diffraction patterns of PTCDI-C_13_ films grown on AlO*_x_* dielectric layer and PI films with different solid contents.

**Figure 4 polymers-15-02421-f004:**
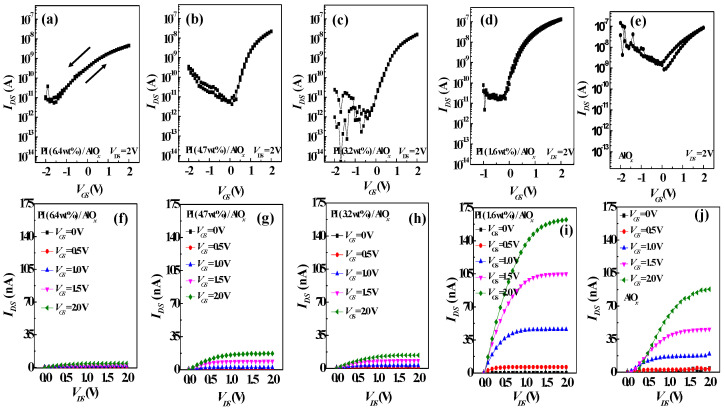
The output and transfer curves of the OFETs with (**a**,**f**) the solid content of PI being 6.4 wt%, (**b**,**g**) the solid content of PI being 4.7 wt%, (**c**,**h**) the solid content of PI being 3.2 wt%, (**d**,**i**) the solid content of PI being 1.6 wt%, and (**e**,**j**) AlO*_x_* dielectric only.

**Figure 5 polymers-15-02421-f005:**
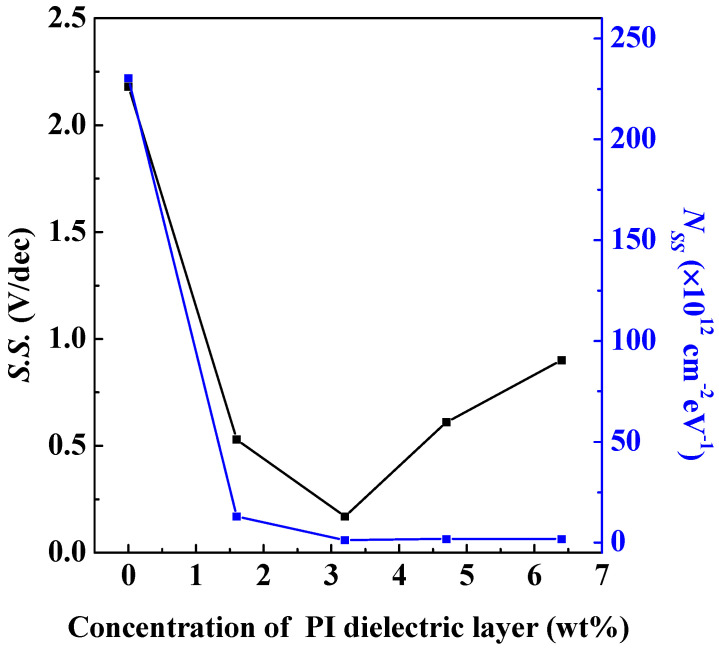
The relationship between the *S.S.* and *N_SS_* of the OFETs versus the different solid contents in the PI layer.

**Figure 6 polymers-15-02421-f006:**
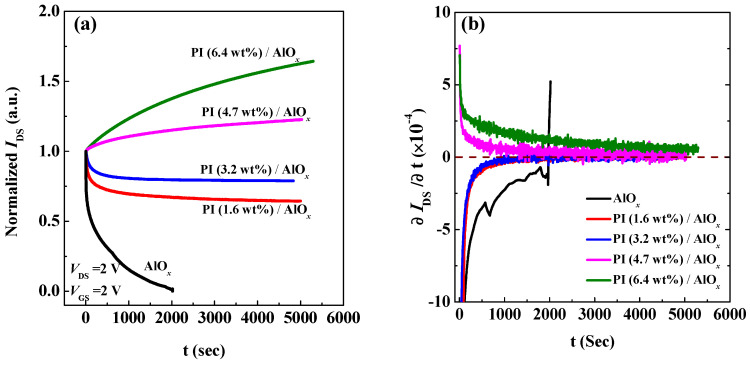
Stability analyses of organic field-effect transistors with different dielectric layers. (**a**) Normalization of the drain current in time; (**b**) differentiation of the drain current with respect to time.

**Figure 7 polymers-15-02421-f007:**
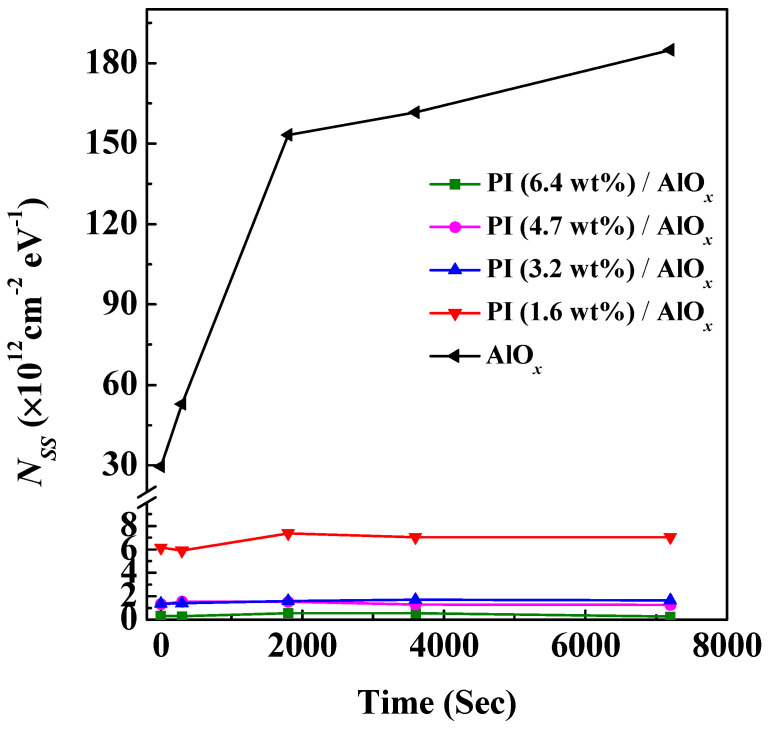
The plot of defect density and time variation for organic field-effect transistors with different dielectric layers.

**Figure 8 polymers-15-02421-f008:**
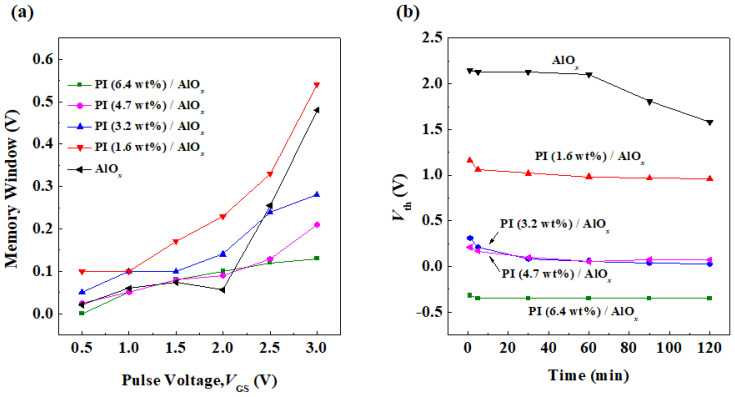
(**a**) Memory window changes under different gate pulse voltages and (**b**) long-term operation threshold voltage changes of organic memory devices with different dielectric layer films.

**Figure 9 polymers-15-02421-f009:**
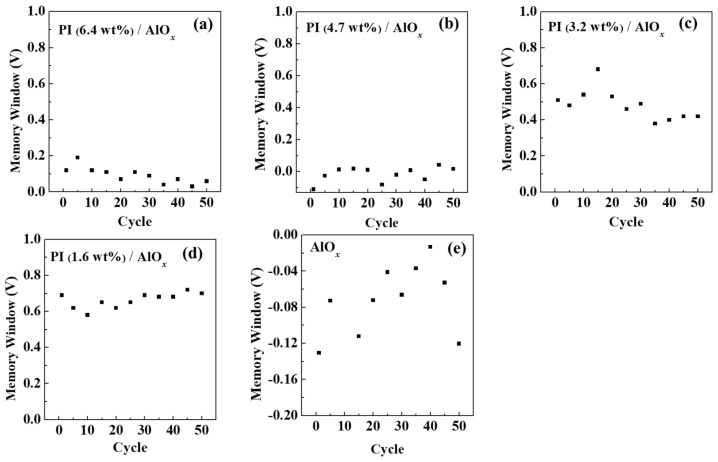
Memory endurance analysis chart of organic memory devices fabricated by using different solid contents of PI films. (**a**) AlO*_x_*/PI (6.4 wt%), (**b**) AlO*_x_*/PI (4.7 wt%), (**c**) AlO*_x_*/PI (3.2 wt%), (**d**) AlO*_x_*/PI (1.6 wt%), and (**e**) native AlO*_x_*.

**Table 1 polymers-15-02421-t001:** Performance criteria of OFETs with different dielectrics.

Dielectric of OFETs	*V*_th_ (V)	*I*_On_/*I*_Off_ Ratio	*C* $\left(\mathrm{nF}/{cm}^{2}\right)$	Mobility $\left({cm}^{2}/Vs\right)$
PI (6.4 wt%)/AlO*_x_*	−0.67	7.0 × 10^2^	19.67	3.22 × 10^−3^
PI (4.7 wt%)/AlO*_x_*	−0.43	4.0 × 10^3^	27.91	3.46 × 10^−3^
PI (3.2 wt%)/AlO*_x_*	0.03	9.2 × 10^3^	72.91	1.52 × 10^−3^
PI (1.6 wt%)/AlO*_x_*	0.05	6.5 × 10^3^	234.41	1.58 × 10^−2^
AlO*_x_*	0.09	4.8 × 10^1^	1007.10	2.29 × 10^−3^

**Table 2 polymers-15-02421-t002:** The parameter of storage carrier density after 50 cycles of each memory device.

Dielectric of OFETs	Cycle Time	*C* $\left(\mathrm{nF}/{cm}^{2}\right)$	Memory Window (V)	Δ*n* (×10^12^ cm^−2^)
PI (6.4 wt%)/AlO*_x_*	50	19.67	0.06	0.007
PI (4.7 wt%)/AlO*_x_*	50	27.91	0.02	0.003
PI (3.2 wt%)/AlO*_x_*	50	72.91	0.42	0.19
PI (1.6 wt%)/AlO*_x_*	50	234.41	0.70	1.02
AlO*_x_*	50	1007.10	−0.12	−0.75

## Data Availability

Data will be made available upon request.

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
