# Peer review of "Enhancement of Stability in n-Channel OFETs by Modulating Polymeric Dielectric"

_polymers, 2023, doi:10.3390/polym15112421_

Round 1
Reviewer 1 Report
Dear Editor,
I went through the paper from Fang and co-workers, about the employment of a double-layer dielectric to improve the stability of n-type OFET.
Although the combination of materials seems innovative, the paper is not suited to be published on Polymers.
The main issue is related to the fact that the authors developed their devices of rigid glass substrates, and that the fabrication process involves high temperature passages (220°C for polyimide curing). At the state-of-the-art, the main endpoint of organic electronics lays in flexible applications, and so the results reported are not competitive with similar studies carried on solution-processable dielectrics and assessment of interface properties carried out on flexible substrates. In conclusion, authors state that proposed devices “will meet consumer demand for flexible electronics products in the future”, but the transfer of their approach to a flexible substrate is not trivial, as the morphology of different layers can significantly change.
A suggestion for the authors is to repeat their research on flexible substrates: for instance, using polyimide substrate would make possible a direct transfer of their technology (without changing fabrication protocols). Although, in that case, the fabrication process would result limited to the employment of thermoplastics as substrates, it would be more appealing for readers interested in flexible electronics applications.
Aside from this consideration, the authors should significantly improve the quality of the presentation.
Title, abstract and introduction are misleading with respect to the focus of the study. The title suggests that the aim is to improve the “stability” of OFET performances, while in the abstract the idea is that the main endpoint is to control memory performances of the device. Also the introduction is not perfectly focused. Finally, the meaning of “stability” must be contextualized, as it may suggest different aspects (stability with respect to bias stress, stability in time). My suggestion is to focus alternatively on the control of memory performance of OFET through the control of the semiconductor/insulator interface, or to focus on the improvement of stability and keeping the characterization of the memory as a benchmark.
It is not clear from the paper how many devices has been tested for each dielectric configuration. Data on Table 1 should be reported in terms of average and standard deviation. The same should be done for subthreshold slope/Nss on Figure 5.
Data showed in Figure 6 does not support the idea of a high stability. Following the results, authors should have targeted a PI concentration in between 3.2wt% and 4.7wt% in order to optimize stability with respect to a continuous bias. In conclusions, authors state that the increase of current in time represents a “excellent stability”, but any significant performance variation can be detrimental for the actual employment of OFETs inside a circuit. An increase of output current of 50% after 1.5h can’t be defined as an excellent result.
As regards the memory characterization:
-
the effect of memory writing-erasing must be showed in terms of shift in the transfer characteristic curve on a device for each dielectric configuration;
-
Endurance tests should be shown in a more representative way if ON and OFF state (in current or in threshold voltage values) are shown, instead of reporting the amplitude of the memory window. The comment about stability of the memory window is also not supported by data: a certain variability is shown also for PI (and it seems similar to the one of native AlOx). It is not explained the reason why the increasing of the solid content of PI reduces the memory window. Moreover, 50 cycles are not enough to demonstrate the endurance, in literature a few hundreds of cycles is generally considered the entry level.
-
Memory performances should also be tested in time, by evaluating in several days/weeks at least that the written state is maintained.
-
An indication on the number of devices tested is important also for this characterization.
Last, but not least, authors should carefully revised the English.
On these bases, I suggest the REJECTION of the paper.
A careful check of the language is needed.
Reviewer 2 Report
In this manuscript, polyimide (PI) was employed to modify the surface of aluminum oxide (AlOx) to serve as the dielectric layer in organic field-effect transistors (OFETs). By utilizing PI solutions with varying solid contents, the properties and reduced trap state density of PI films were adjusted to achieve controlled stability in N, N'-11 ditridecylperylene-3,4,9,10-tetracarboxylic diimide (PTCDI-C13)-based OFETs. Additionally, the OFET-based memory devices featuring PI films demonstrated not only remarkable memory retention but also consistent memory durability. These results are very interesting but there are some minor issues in the manuscript. It is suggested to be accepted after minor revision based on the following comments.
- The authors need to include the chemical structures and sources for both polyimide and PTCDI-C13.
- Although the results are divided into several sections, the connections between them need to be strengthened. As it currently stands, the manuscript reads more like a laboratory report.
- It is recommended that raw data be incorporated to illustrate the threshold voltage in the write-erase cycle.
The manuscript's readability would benefit from language refinements, ensuring a smoother flow. Additionally, enhancing the coherence between various results sections will further improve the overall presentation.
Reviewer 3 Report
In this article, the authors present work on the OFET and memory devices using different dielectric layers. The article will be a good reference for researchers working on the topic. Reviewer has the following recommendations.
1. Please use better readable images particularly Fig. 4 and Fig 9.
For fig. 2 provide the high-resolution images, for fig, 3 zoom in the actual XRD peaks.
2. Recommended to provide the actual thickness of the dielectric layers. For a better comparison it would be good to have the same thickness for all the different dielectric layers (e.g. with different PI Wt% ). And then for a particular wt% to study the effect of the variation with the thickness.
3. For line 216 to 220 author mentions comparison however comparative values or discussions is missing. In the results and discussion references are missing, please provide the references for your explanation for the phenomenon or the effects For e.g. for their discussion on trapping
4. From Fig. 2 and Fig. 3 the PI 6.4wt% has a smoother surface and better crystallinity, so it is expected that the organic grown on them has higher performance and better mobilities however it is not the case, please provide the additional specific discussion on that.
5. For the memory characteristics please provide the actual characteristic and the device and the writing cycles. Following ref. might be helpful /https://onlinelibrary.wiley.com/doi/full/10.1002/inf2.12186
Please check spellings and grammar
Author Response
Please see the attached manuscript.

Round 2
Reviewer 1 Report
Dear Editor,
I went through the revised version of the manuscript by Fang and co-workers, and I also considered their response to my remarks.
Unfortunately, I didn't found any of the main point properly addressed:
1) the Reviewer understand that the choice for a rigid substrate is justified by the need of ideal interfaces to study the effect of wt% of PI in device performances; nonetheless, these results cannot be easily sold as remarkable for flexible electronics applications;
2) the authors still claim that having inverted the current shift direction justifies the "excellent stability". They can surely conclude that they are fixing the problem of bias stress, but "current stability" means that the current only slightly changes in time. In their results, the best performance in terms of stability has been obtained indeed for 3.2wt% of PI, with a current decrease of about 20% in less than two hours operation. When bias stress is avoided, current increases are even worst, and its stabilization is also less evident (with bias stress, a plateau was reached). Both current decrease and increase are detrimental for circuit stability, so their claim is not supported by facts.
3) Although authors replied that 50 cycles are enough to demonstrate endurance, literature demonstrates that a few hundreds are the standard for a proper memory characterization. Moreover, "memory window" is not totally representative of the memory performance, as its reduction cannot be directly related to an increase of the OFF current or a decrease of the ON current. Finally, retention time should be also taken into account.
On these bases, I confirm my suggestion of REJECTION of the paper.
Some typos and minor grammar errors still need to be solved.
